# Bacterial Resistance in Pneumonia in Developing Countries—A Role for Iron Chelation

**DOI:** 10.3390/tropicalmed4020059

**Published:** 2019-04-10

**Authors:** Sufia Islam, Mohammod Jobayer Chisti, Muniruddin Ahmed, Nafiza Anwar, Christian Lehmann

**Affiliations:** 1Department of Pharmacy, East West University, Dhaka 1212, Bangladesh; 2Centre for Nutrition & Food Security, International Centre for Diarrhoeal, Disease Research, Dhaka 1213, Bangladesh; chisti@icddrb.org; 3Department of Clinical Pharmacy and Pharmacology, University of Dhaka, Dhaka 1000, Bangladesh; drmuniruddin@gmail.com; 4Royal College Affiliated Hospital for Education and Training Institute, Dhaka 1212, Bangladesh; nafiza_anwar@hotmail.co.uk; 5Department of Anesthesia, Pain Management and Perioperative Medicine, Faculty of Medicine, Dalhousie University, Halifax, NS B3H 1X5, Canada; chlehmann@dal.ca

**Keywords:** pneumonia, iron, iron chelation, resistance, antibiotics, sepsis

## Abstract

Pneumonia represents one of the major infectious diseases in developing countries and is associated with high mortality, in particular in children under the age of five. The main causative bacterial agents are *Streptococcus pneumoniae* and *Haemophilus influenzae* type B, accounting for 33% and 16%, respectively, of the mortality in under-fives. Iron modulates the immune response in infectious diseases and increased iron levels can lead to complications such as sepsis and multiorgan failure. This review will look into the use of iron chelators in order to reduce microbial growth and attenuate a dysregulated immune response during infection. Our hypothesis is that temporary restriction of iron will lessen the incidence and complication rate of infections like pneumonia and result in a decrease of mortality and morbidity.

## 1. Introduction

Pneumonia is recognized as one of the leading causes of mortality in developing countries, in particular in children under the age of five [1]. The most important causative bacterial pathogens for childhood pneumonia are *Streptococcus pneumoniae* and *Haemophilus influenzae* type B. Prior to the introduction of conjugate vaccines in lower and middle-income countries (LMIC), 33% and 16%, respectively, of deaths were caused by *S. pneumoniae* and *H. influenzae* type B [2]. A major problem in health care is the development of resistance to current antibiotic treatments [3]. The ongoing rapid spread of resistance mechanisms has led to the development of multidrug resistance strains, thereby limiting the use of antimicrobials [4]. Sepsis is the terminal event in acutely ill children with pneumonia leading to severe sepsis and septic shock [5]. Invading microorganisms require iron for their multiplication. They utilize siderophores (natural iron chelators) to obtain iron from the host. On the other hand, the host forms superoxide radicals using Fe^2+^ in the Fenton and Haber–Weiss reactions to be released during the immune response to microbial invasion. However, in order to reduce the iron availability for microbes, the organism restricts iron during infection. Therefore, the iron status is an important factor in the fight against pulmonary infections [6].

## 2. Epidemiology

In the developing world, pneumonia is not only more common than it is in Europe and North America, but it is also more severe and represents the disease with the highest mortality rate in children. Pneumonia is currently the leading cause of death among children under the age of 5 years worldwide, being responsible for more than 900,000 of the annual 6.3 million deaths in this age group and with more than 95% of these deaths occurring in developing countries [7,8,9,10]. Almost 70% of the cases come from Africa and Southeast Asia, with one child dying every minute [11]. In Bangladesh, among the 129,000 children under the age of five who die, 13% die from pneumonia [7]. According to recent data from 89 eligible studies in 2010, 11.9 million episodes of severe and three million episodes of very severe acute lower respiratory tract infections (ALRI) resulted in hospital admissions in young children worldwide [10]. The incidence was higher in boys than in girls, with the sex disparity being greatest in South Asian studies [9]. In a study conducted at Dhaka Hospital among 401 children under five with ALRI, it was observed that the most common manifestation was pneumonia, and a respiratory pathogen (both bacterial and viral) was identified in 30% of cases. The case fatality rates were 14% in bacterial pneumonia and 3% in viral pneumonia [12]. Pneumonia is a major cause of health care utilization, hospitalization, and death in children in developing countries [13]. Therefore, improvement in the case-management strategies of the major causes of child death, such as pneumonia and neonatal illnesses, should be a priority to improve child survival in developing countries. 

## 3. Resistance

Antibiotic resistance has been rising rapidly over the years. This has been a matter of concern from the onset of modern-day antibiotic use around 75 years ago [14]. Alexander Fleming highlighted his concerns about antibiotic resistance in 1945 in his Nobel lecture: “There is the danger that the ignorant man may easily underdose himself by exposing his microbes to non-lethal quantities of the drug and make them resistant” [15]. These same concerns exist now and can be linked to the overuse and misuse of antibiotics in clinical practice [16]. It has been reported that the use of antibiotics in animals bred for human consumption also contributes to antibiotic resistance, and this is somewhat underrated. The outcome of this is global and more apparent in various regions and health institutions [14]. 

Antibiotic resistance is of grave importance and should be high on the agenda on the list of global problems. It has been recently stated that the efficacy of antibiotics has been reduced and consequently those drugs fail to work against infections. The authors looked at 12 commonly used antibiotics and through the method of kinetic growth assays identified that an extended lag phase offers bacterial advantages and enhances regrowth once the antibiotic has been removed. This finding is of significant value in clinical practice, as bacterial resistance is often underestimated, leading to the re-emergence of the condition [17].

Many infections will become immensely difficult to treat, for example infections caused by carbapenemase-producing *Klebsiella pneumonia* (KP), which has been found to be resistant to antibiotics like Tigecycline, Colistin and aminoglycosides. This is a serious matter when faced with infections that are difficult to manage, as the options for treatment become limited [18]. A very recent study from China also reported that among 444 multi-drug resistant KP, 299 strains were extended-spectrum β-lactamase (ESBL) KP. The report highlighted that the prevalence and resistant rates of ESBL KP have gradually increased in ICU patients in the Republic of China in the last 5 years [19].

Penicillin resistance was initially identified in 1967. The Asian Network for Surveillance of Resistance Pathogens (ANSORP) conducted a study between 2000 and 2001 and found that there were high rates of penicillin resistance in many Asian countries, with the highest being in Vietnam (71.4%) and lowest in Taiwan (38.6%) [20,21]. The main mechanism responsible for resistance in Penicillin is alteration of the cell wall penicillin-binding proteins which bring about a decreased affinity for penicillin [22]. A multinational surveillance study found that many Asian countries indicated that there was a distinct rise in the prevalence rates and the levels of antimicrobial resistance among *S. pneumoniae* isolates. Fluoroquinolones such as Moxifloxacin and Gatifloxacin have been found to be resistant to *S. pneumoniae* [20]. This matter again highlights that there is an urgent need to improve infection control and control how antibiotics are used to maximize their effect without compromising the patient and society as a whole.

The findings from the Felmingham (2007) study conclude that, as previously suggested, there is a high prevalence of resistance to commonly used antibiotics. The study also identified multiple resistance phenotypes in *S. pnuemoniae*. The results indicate that high-level macrolide resistance is still on the rise on a global scale. Ribosomal methylation mediated by erm(B) is the most common mechanism of macrolide resistance. However, in other countries such as Canada, Greece and the USA, the drug efflux mediated by mef(A) was pronounced [23]. 

A recent study identified which antibiotics have been shown to be ineffective in the treatment of Group B streptococcal (GBS) pneumonia over the last 10 years. Over the years, GBS has become resistant to many antibiotics, evoking worldwide interest. In conclusion, patients with GBS pneumonia exhibited antibiotic resistance to the following antibiotics: gyrase inhibitors, macrolides, co-trimoxazole, lincosamides, and aminoglycosides [24].

It is a moral obligation for scientists and medical professionals to try to control and slow down the process of antibiotic resistance. If the problem persists, we will see a rise in infection rates as the emergence of hard-to-treat infections will prevail, with a transition to the pre-antibiotic days. This consequently will cause a rise in morbidity and mortality, and a major setback to society as a whole.

## 4. Iron Chelation

Iron represents an essential trace element for the proper functioning of all living cells. This metal is important for a wide variety of cellular events. For example, iron plays an integral role in electron transfer, a process where reactive oxygen species (ROS) are generated. Superoxide radicals can cause cell damage and result in apoptosis. Therefore, iron levels in the body need to be tightly controlled [25]. Iron is stored in hepatocytes and macrophages in the liver and spleen. Hepcidin, a hormone produced by the liver, is the main regulator of systemic iron levels [6]. Iron requirements are high during infancy, childhood and pregnancy. Ferric iron in the diet is converted to ferrous iron by duodenal cytochrome b. There are two fates of iron according to the body’s requirements. Iron not immediately required by the body is stored within ferritin. When iron demand is high in the body, it is exported into the circulation via ferroportin1 and ultimately binds to transferrin [26,27].

Microorganisms also need iron for several vital functions. Of particular interest in the context of pneumonia is their iron consumption from the hosts for their multiplication. For example, proliferation was reduced in *Mycobacterium tuberculosis* infection by iron withdrawal [28]. Another more recent study showed that animals infected with siderophore-deficient *M. tuberculosis* have significantly reduced growth when compared to animals infected with wild-type bacteria [29]. Iron chelation in combination with vancomycin treatment was also effective in methicillin-resistant *S. aureus* (MRSA) infected mice [30]. Multi-drug resistant bacteria (MDRB) can be life-threatening for hospitalized patients. Long hospitalization, severe complications and decreased immunity render in particular ICU patients more susceptible to infection [31]. Therefore, iron restriction represents a potential additional defense mechanism in the host. Evidence suggests that an overload of iron debilitates the phagocytic activity to destroy microorganisms [25]. Iron chelation therapy has been proven to have a positive outcome in several types of infection. However, as stated by Kontoghiorghes et al., deferoxamine (DFO) is able to act as a siderophore for the microbes and exacerbate some infections [32], whereas other chelators like deferiprone have the highest therapeutic index for long-term antimicrobial activity. 

## 5. Conclusions

Iron chelators, when used therapeutically, can reduce the growth of microorganisms and potentiate antimicrobial strategies in pneumonia. In situations where antimicrobial treatment has been unsuccessful or where current therapies have caused resistance, iron chelation should be considered after the confirmation of efficacy in randomized controlled trials.

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
