# Peer review of "Bacterial Resistance in Pneumonia in Developing Countries—A Role for Iron Chelation"

_tropicalmed, 2019, doi:10.3390/tropicalmed4020059_

Round 1

Reviewer 1 Report

This is a short paper about an important topic. The focus is on developing a hypothesis that restriction of iron is an approach to be used in treatment of infections, particularly when used in combination with antibiotic treatment. Overall, the manuscript is well written and understandable. Minor revision, particularly the occasional need for a comma, would be helpful.

Author Response

Reviewer 1:

Comment: This is a short paper about an important topic. The focus is on developing a hypothesis that restriction of iron is an approach to be used in treatment of infections, particularly when used in combination with antibiotic treatment. Overall, the manuscript is well written and understandable. Minor revision, particularly the occasional need for a comma, would be helpful.

Response: Thank you very much for the kind comments on our work titled “Bacterial resistance in pneumonia in developing countries – a role for iron chelation”. Some minor revision, particularly the occasional need for a 

comma has been done to increase the quality of the manuscript.

Reviewer 2 Report

Comments: This reviewed  on bacterial resistance in pneumonia in developing countries – a role for iron chelation, showed that chelation therapy has been proven to have a positive outcome in several types of infection. Deferiprone appears to have the highest therapeutic index for antimicrobial activity 

Antibiotic resistance of bacteria associated with other causes,  is the main cause of non-effective therapy of nosocomial infections and sepsis. As we can see antibiotic resistance of K. pneumoniae strains is associated mainly with the production of ESBL 

There was no significant increase in the frequency or duration of respiratory and enteric infections after 12 months of iron supplementation.

In a recent prospective randomized study of prepubescent schoolchildren in Papua New Guinea treated with oral iron, there was no difference in parasite infections rate, parasite density, or levels of antimalarial immunoglobulin between children receiving oral iron and the control group. 

This review shows significant results and deserves publication after minor  revisions. It emphasizes that future research may lead to the identification of iron chelators with considerable usefulness in the control of infectious disease. 

Author Response

Reviewer 2:

Comment: This review on bacterial resistance in pneumonia in developing countries – a role for iron chelation, showed that chelation therapy has been proven to have a positive outcome in several types of infection. Deferiprone appears to have the highest therapeutic index for antimicrobial activity. Antibiotic resistance of bacteria associated with other causes, is the main cause of non-effective therapy of nosocomial infections and sepsis. As we can see antibiotic resistance of K. pneumoniae strains is associated mainly with the production of ESBL. 

Response:  We agree with the reviewer that for nosocomial infections and sepsis effective therapy is yet to be found. We have pointed out the increased prevalence of ESBL producing K. pneumoniae strains in ICU patents in China (Line 84 to 86) and added the following reference:

Wang C, Yuan 1, Huang W, Yan L, Tang J LC. Epidemiologic analysis and control strategy of Klebsiella pneumoniae infection in intensive care units in a teaching hospital of People ’ s Republic of China. Infect Drug Resist. 2019;12:391–8.

Comment: There was no significant increase in the frequency or duration of respiratory and enteric infections after 12 months of iron supplementation. In a recent prospective randomized study of prepubescent schoolchildren in Papua New Guinea treated with oral iron, there was no difference in parasite infections rate, parasite density, or levels of antimalarial immunoglobulin between children receiving oral iron and the control group. 

Response:  We agree – in some iron supplementation studies, there was no significant increase in side effects such as enteric or respiratory infections. However, we propose to utilize the potential benefits of iron chelation during infection, rather than a general avoidance of iron supplementation, which might be indicated for other reasons.

Comment: This review shows significant results and deserves publication after minor revisions. It emphasizes that future research may lead to the identification of iron chelators with considerable usefulness in the control of infectious disease. 

Response: Thanks for the comment. Indeed, future studies are needed to identify the role of iron chelators to control infection, thus reduce the morbidity and mortality specially in children from developing countries.